# FDA: Generating Fair Synthetic Data with Provable Trade-off between Fairness and Faithfulness

## Abstract

We propose a novel framework called FDA for generating Fair synthetic data through Data Augmentation, *offering the first method with provable trade-off guarantee between fairness and faithfulness*. Unlike other existing methods, our approach utilizes a novel joint model that consists of two sub-models: one focused on enforcing strict fairness constraints while the other dedicated to preserving fidelity to the original data, coupled with a tuning mechanism that provides explicit control over the trade-off between fairness and faithfulness. Specifically, our FDA framework enables explicit quantification of the extent to which the generated fair synthetic data preserve faithfulness to the original data, while achieving an intermediate level of fairness determined by a user specified parameter $\alpha \in [0, 1]$. Theoretically, we show that the resulting fair synthetic data converge to the original data in probability when $\alpha$ tends to 1, thereby implying convergence in distribution. Our framework can be also combined with some GAN-based fair models, such as DECAF, to further improve the utility of the resulting synthetic data in downstream analysis, while carefully balancing fairness. Furthermore, we obtain an upper bound of the unfairness measurement for downstream models trained on the generated fair synthetic data, which can help users to choose appropriate $\alpha$. Finally, we perform numerical experiments on benchmark data to validate our theoretical contributions and to compare our FDA with other methods.

## 1 Introduction

Artificial intelligence (AI) and machine learning (ML) algorithms have increasingly been used to improve decision-making in almost all aspects of our lives (Zhao et al., 2018; Bogen & Rieke, 2018; Cohen et al., 2020; Mukerjee et al., 2002; Angwin et al., 2016; Berk et al., 2021). However, there is mounting evidence showing that the developed algorithms may inherit biases and injustices from historical data, leading to unfair decisions that discriminate against certain populations (Dastin, 2018; Datta et al., 2018; Lu et al., 2020; de Vassimon Manela et al., 2021).If not properly addressed, biased or unfair decision-making may lead to violations of equality and anti-discrimination laws (Krishnamurthy, 2021; Wachter et al., 2021). The emerging field of algorithmic fairness seeks to address this urgent issue by mitigating the bias and discrimination in the AL and ML systems.

Broadly speaking, the bias mitigation methods can be categorized into three types: pre-processing, in-processing, and post-processing. For a comprehensive overview of these methods, we encourage readers to refer to recent review papers, such as (Pessach & Shmueli, 2022; Hort et al., 2022; Mehrabi et al., 2021; Caton & Haas, 2024), and the extensive references cited within these works. Pre-processing methods modify the biased training data, with the goal that any downstream model trained on debiased data would achieve desired fairness requirements. In-processing methods modify the algorithms by enforcing fairness constraints during training, with the goal that the trained algorithms achieve the desired fairness requirements on all real-life data. Post-processing methods modify the predictions based on a trained unfair model, with the goal that the final predictions satisfy certain fairness requirements. In recent years, as a pre-processing method, fair synthetic data generation has gained significant momentum (Feldman et al., 2015a; Zhang et al., 2017b; Calmon et al., 2017; Xu et al., 2018; Zemel et al., 2013; Xu et al., 2019; van Breugel et al., 2021; Rajabi & Garibay, 2022). For example, Xu et al. (2019) proposed FairGAN, a GAN-based method to create

Table 1: Overview of related fair synthetic data generation methods summarized according to the following features: **(1)** supports trade-off between fairness and data utility; **(2)** allows continuous labels; **(3)** allows categorical labels; **(4)** provides explicit quantification of loss of faithfulness to the original data, when meeting user-specified fairness requirement; **(5)** provides theoretical guarantee on the convergence of the generated fair synthetic data to the original data in probability and distribution; **(6)** provides theoretical analysis on the fairness of downstream models using debiased synthetic data.

| Model | Reference | (1) | (2) | (3) | (4) | (5) | (6) |
|---|---|---|---|---|---|---|---|
| FDA | THIS PAPER | ✓ | ✓ | ✓ | ✓ | ✓ | ✓ |
| DECAF | VAN BREUGEL ET AL. (2021) | ✗ | ✓ | ✓ | ✗ | ✗ | ✗ |
| FAIRGAN | XU ET AL. (2018) | ✓ | ✓ | ✓ | ✗ | ✗ | ✗ |
| OPPDP | CALMON ET AL. (2017) | ✓ | ✗ | ✓ | ✗ | ✗ | ✗ |
| TABFAIRGAN | RAJABI & GARIBAY (2022) | ✓ | ✗ | ✓ | ✗ | ✗ | ✗ |

synthetic data that satisfy group fairness; van Breugel et al. (2021) proposed DECAF, which trains a graphical causal model using GANs and allows desired fairness constraints imposed via the associated causal graphs. Despite the successes of these earlier works, an important question remains unaddressed: *To what extent does the fair synthetic data represent the statistical properties of the original data, preserving its utility for downstream analysis and modeling?*

Generally, the task of generating a fair synthetic data that preserves the properties of the original data creates a tension. On one hand, achieving fairness requires modification of the unfair data, which may inadvertently impact the faithfulness of the synthetic data to the original data. On the other hand, the synthetic data should faithfully represent the statistical properties of the original data, in order to preserve its utility for downstream analysis and modeling. Therefore, achieving fairness requires sacrificing some level of faithfulness, and vice versa. Due to the inherent competing nature between these two goals, a trade-off between fairness and faithfulness is necessary characterized when generating fair synthetic data. In practice, striking the right balance involves careful consideration of the goals, stakeholders' priorities, and ethical implications in a given application.

**Contribution.** In this paper, we propose FDA, a Fair synthetic data generation framework through Data Augmentation. FDA is built upon a joint modeling framework consisting of a fair model $\mathcal{M}_{\text{fair}}$ and a faithful model $\mathcal{M}_{\text{faithful}}$, coupled with a tuning mechanism to achieve a provable trade-off between fairness and faithfulness in the generated synthetic data. This allows an explicit quantification of the extent to which the generated fair synthetic data preserve faithfulness to the original data, while meeting specific fairness requirement controlled by $\alpha \in [0, 1]$, a user specified bias reduction parameter that quantifies the amount of biases removed from the original unfair data. Theoretically, setting $\alpha = 0$, the resulting synthetic data satisfy absolute fairness, with maximum reduction of the faithfulness to the original data. Conversely, setting $\alpha = 1$, the resulting synthetic data achieves perfect similarity to the original data and is proved to converge to the original data in probability, further implying convergence in distribution. When setting $\alpha \in (0, 1)$, users can achieve an intermediate level of fairness while maintaining a certain level of faithfulness to the original data, both quantified by $\alpha$. Our framework can be combined with GAN-based fair models such as DECAF, to further improve the data utility of the resulting fair synthetic data in downstream analysis, while achieving an intermediate level of fairness. In contrast to black-box methods that require time-consuming training, our FDA framework generates synthetic data directly from the predictive distributions defined by our chosen joint model, which follows simple Gaussian distributions. Furthermore, to guide users to choose appropriate unfairness reduction parameter $\alpha$, we provide theoretical analysis on the fairness of downstream models trained on the generated fair synthetic data. As far as we know, our FDA is the first method to provide all these desired features, with a comparison of our FDA with other methods summarized in Table 1.

**Notations.** For any positive integer $K$, let $[K] = \{1, \cdots, K\}$. For any $p \geq 1$, let $\mathcal{M}_p(\mathbb{R})$ be the space of all probability measures on $\mathbb{R}$ with finite $p$-th moment. For any two random variables $U$ and $V$, $\mu_{U|V}$ denotes the conditional distribution of $U$ given $V$; $\mu_U$ and $\mu_V$ denote their marginal distributions respectively. We write $U \overset{d}{=} V$ when $U$ and $V$ are equal in distribution. For a random sequence $\{U_i\}_{i=1}^{\infty}$, we write $U_n \overset{p}{\to} U$ ($U_n \overset{d}{\to} U$) when $U_n$ converges in probability (in distribution) to $U$ as $n \to \infty$. We use $\Delta^{K-1}$ as the probability simplex in $\mathbb{R}^K$.

## 2 PRELIMINARIES

A sequence of triplets $\mathcal{D} = \{Y_i, X_i, S_i\}_{i=1}^n$ is observed, where for each $i$, $Y_i \in \mathbb{R}$ denotes the outcome, $S_i \in [K]$ denotes the sensitive attribute, and $X_i \in \mathbb{R}^d$ denotes other attributes. Assuming that $\mathcal{D}$ is sampled from the distribution $\mathcal{P}_{\mathcal{D}}$, which violates certain fairness requirements, rendering $\mathcal{D}$ unfair, our objective is to generate fair synthetic data denoted as $\hat{\mathcal{D}}$ based on $\mathcal{D}$. Specifically, we want to ensure that $\hat{\mathcal{D}}$ satisfies $\alpha$-reduction of unfairness (given in Theorem 3.5), for any fixed $\alpha \in [0, 1]$.

As discussed in van Breugel et al. (2021), predictive fairness measures such as equalized odds are not compatible in the context of fair data, as the aim is to ensure the fairness in the synthetic data distribution, rather than achieving fair algorithmic predictions. Consequently, we follow van Breugel et al. (2021) to focus on *Demographic Parity* (DP) and formally extend it to the context of fair data.

**Definition 2.1** (Demographic parity). The distribution $\mathcal{P}_{\mathcal{D}}$, from which $\mathcal{D}$ is sampled, is said to satisfy *demographic parity (DP)*, if it satisfies $(Y|S = s_1) \overset{d}{=} (Y|S = s_2)$, for any $s_1, s_2 \in [K]$. In other words, for any $T \subseteq \mathbb{R}$, $\mathbb{P}(Y \in T|S = s_1) = \mathbb{P}(Y \in T|S = s_2)$.

Then, any discrepancy between the distribution of $Y|S = s_1$ and that of $Y|S = s_2$, for any $s_1, s_2 \in [K]$ indicates violation of DP. Note, our proposed framework can be applied to the conditional fairness notion (Barocas et al., 2023) (see Remark 3.2). As discussed in Chzhen & Schreuder (2022), various distance measures, e.g., Wasserstein distance, total variation and Kolmogorov-Smirnov distance, have been used to evaluate this discrepancy empirically and thereby quantify the violation of DP. In this paper, we follow Chzhen & Schreuder (2022) and others (Gouic et al., 2020; Chzhen et al., 2020; Jiang et al., 2020; Gaucher et al., 2022; Xian et al., 2022) to use Wasserstein distance due to its effectiveness to explicitly quantify unfairness measurement, faithfulness to the original data as well as data utility in downstream models using the same unit measurements comparing to other distance measures. Specifically, we define the unfairness measure of the original distribution of $\mathcal{D}$ as the sum of the weighted distances between $\{\mu_{Y|S=s}\}_{s \in [K]}$ and their common barycenter (Villani, 2021; Santambrogio, 2015), *w.r.t.* the Wasserstein-2 distance[1], denoted by $W_2(\mu_{Y|S=s}, \cdot)$, as defined below.

**Definition 2.2** (Unfairness measure). We define the *unfairness* of the distribution $\mathcal{P}_{\mathcal{D}}$, from which the dataset $\mathcal{D}$ is sampled, as follows [2]

$$\mathcal{UF}(\mathcal{P}_{\mathcal{D}}) := \min_{\nu \in \mathcal{M}_2(\mathbb{R})} \sum_{s=1}^{K} \omega_s W_2(\mu_{Y|S=s}, \nu), \tag{1}$$

for any given weights[3] $(\omega_1, \cdots, \omega_K) \in \Delta^{K-1}$.

It is easy to see that $\mathcal{UF}(\mathcal{P}_{\mathcal{D}}) = 0$ if and only if there is a minimizer $\nu$ in equation 1 such that $\mu_{Y|S=s} = \nu$ for all $s \in [K]$, that is, it satisfies the DP constraint: $(Y|S = s_1) \overset{d}{=} (Y|S = s_2)$ for all $s_1, s_2 \in [K]$. Conversely, a larger value of this measure[4] indicates a more severe violation of DP constraint.

**Problem statement.** For any biased dataset $\mathcal{D}$, and a user-specified *bias reduction factor* $\alpha \in [0, 1]$, we substitute the observed $Y_i$ values with their synthetic counterparts to produce fair synthetic data, denoted as $\hat{\mathcal{D}} = \{\hat{Y}_i, X_i, S_i\}_{i=1}^n$. Here $\hat{\mathcal{D}}$ satisfies the following bias reduction guarantee:

- $\mathcal{UF}(\mathcal{P}_{\hat{\mathcal{D}}}) = \alpha \mathcal{UF}(\mathcal{P}_{\mathcal{D}})$ (Theorem 3.5). where $\mathcal{P}_{\hat{\mathcal{D}}}$ denotes the distribution of $\hat{\mathcal{D}}$.

In the meanwhile, we can assess the loss of faithfulness between the synthetic data $\hat{\mathcal{D}}$ and the original data $\mathcal{D}$ by calculating Wasserstein-2 distance between $\mu_{\hat{Y}}$ and $\mu_Y$, which is a closed-form function of $\alpha$ (Theorem 3.6). These findings enable users to choose an appropriate $\alpha$ by considering the explicit trade-off between fairness and faithfulness.

---

[1]See Section A.1 for a formal definition of Wasserstein-$p$ distance.

[2]One may use $W_2^2(\mu_{Y|S=s}, \nu)$ when defining $\mathcal{UF}(\mathcal{P}_{\mathcal{D}})$; then $\alpha$ needs to be replaced by $\alpha^2$.

[3]See Appendix A.3 for a discussion on how to select the weights.

[4]See Section A.2 for a discussion on how to evaluate this unfairness measure empirically.

Importantly, under our FDA framework, we can generate high quality synthetic data with a theoretical guarantee that as $\alpha$ approaches 1, the synthetic $\hat{Y}$ converges to the original $Y$ in probability (and consequently in distribution) conditional on the features $X, S$.

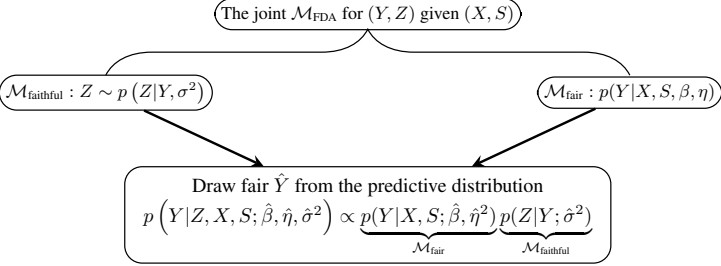

Figure 1: A graphical representation of our FDA synthetic data generation framework.

## 3 THE PROPOSED FDA METHOD

**Overview.** The main idea is to simulate synthetic data from the predictive distribution defined under a joint model, denoted as $\mathcal{M}_{\text{FDA}}$, using both observed $\mathcal{D}$ and additionally augmented data. A similar approach was previously considered in Jiang et al. (2022) to generate privacy preserving synthetic data. In this work, we build upon their method to address the challenge of generating fair synthetic data, providing provable theoretical guarantees on the trade-off between fairness and utility, as well as the convergence of the synthetic data to the original data in both probability and distribution.

Specifically, $\mathcal{M}_{\text{FDA}}$ consists of two sub-models: (i) a fair model (see equation 2), denoted as $\mathcal{M}_{\text{fair}}$, which specifies certain relationship between $Y_i$ and the feature vector $X_i$ and the sensitive attribute $S_i$, such that it imposes exact fairness constraint; and (ii) a faithful model, $\mathcal{M}_{\text{faithful}}$, which generates $Z_i$ (see equation 3 for details) given $Y_i$, for each $i \in [n]$, such that $Z_i$ are noisy copies of $Y_i$ with accuracy level controlled by tuning parameters. The fair synthetic data are then generated as samples from the corresponding predictive distributions that are defined by the model $\mathcal{M}_{\text{FDA}}$.

By design, these $Z_i$ contain information about $Y_i$ so that $\mathcal{M}_{\text{faithful}}$ plays the role of enforcing the resulting synthetic data to be close (and thus faithful) to the original data. In contrast, $\mathcal{M}_{\text{fair}}$ plays the role of imposing the desired fairness requirement, e.g., the DP constraint. Under such a framework, both models $\mathcal{M}_{\text{faithful}}$ and $\mathcal{M}_{\text{fair}}$ influence the resulting synthetic data, with their respective contributions determined by the values of the tuning parameters introduced in $\mathcal{M}_{\text{faithful}}$ (can be seen in equation 3 and discussed thereafter). As a result, the tuning parameters control the relative influence from $\mathcal{M}_{\text{faithful}}$ and $\mathcal{M}_{\text{fair}}$, and thus balance between the two competing goals, fairness and faithfulness to the original data. Given the proposed framework is general, the data considered can be any type and need not to be limited to be binary or discrete as considered by many others. Next, we present details of $\mathcal{M}_{\text{faithful}}$ and $\mathcal{M}_{\text{fair}}$ when $Y_i$'s are continuous; the discussion on how to use the proposed framework when $Y_i$'s are categorical is given in Remark 3.4.

**Fair model.** The fair model $\mathcal{M}_{\text{fair}}$ can be written in the following general form:

$$\mathcal{M}_{\text{fair}} : Y_i = f(X_i, S_i, \beta) + \varepsilon_i, \ \varepsilon_i \overset{iid}{\sim} \mathcal{N}(0, \eta^2), \tag{2}$$

where $f$ can be any fair predictor, under which $Y_i$ satisfies the DP constraint, and $\beta$ is the associated model parameter. Notably, the simplest choice of $\mathcal{M}_{\text{fair}}$ is a constant mean model (CMM) with $f = \beta_0$ being a constant function. Alternatively, one could also choose $f$ to be a GAN-based fair predictor, e.g., DECAF (discussed in Remark 3.3).

However, since many dependencies among the variables in the original data $\mathcal{D}$ are intentionally omitted in $\mathcal{M}_{\text{fair}}$ to fulfill the fairness requirement, loss of information and utility is inevitable. In other words, generating synthetic data solely from $\mathcal{M}_{\text{fair}}$ could result in considerable loss of faithfulness to the original data. As pointed out by many others (Feldman et al., 2015a; Xu et al., 2018; Chzhen & Schreuder, 2022; Zhao & Gordon, 2022; Tran et al., 2022), a sustainable solution would be to accept some degree of compromise on fairness in order to preserve the utility of the original data in the downstream analysis. We achieve this goal by introducing a faithful model as a

submodel in our joint modeling framework, which plays the role of mitigating loss of information about the original data.

**Faithful model.** The faithful model $\mathcal{M}_{\text{faithful}}$ takes the following form:

$$\mathcal{M}_{\text{faithful}} : Z_{im} = Y_i + e_{im}, \ e_{im} \overset{iid}{\sim} \mathcal{N}(0, \sigma^2), \tag{3}$$

for $m \in [M]$. The specification of this model allows us to generate noisy copies of $Y_i$, i.e., $Z_i = (Z_{i1}, \cdots, Z_{iM})^\top$, so that $Z_i$ contains information about $Y_i$ with their faithfulness to the original $Y_i$ controlled by the tuning parameters $\sigma^2$ and $M$. By increasing the number of copies, $M$ and/or decreasing the additive noise variance $\sigma^2$, the generated $Z_i$ contains more information about $Y_i$. In practice, one can fix the value of $M$ and adjust the value of $\sigma^2$, or vice versa. As discussed next in Remark 3.1, the ratio $\sigma^2/M$ determines both the levels of fairness in the generated synthetic data as well as its faithfulness to the original data.

With $\mathcal{M}_{\text{fair}}$ and $\mathcal{M}_{\text{faithful}}$ chosen, we create our augmented dataset $\mathcal{D}_{\text{DA}} = \{\mathcal{D}, \{Z_i\}_{i=1}^n\}$ given the simulated $Z_i$'s, and fit our joint model $\mathcal{M}_{\text{FDA}}$ given below,

$$\underbrace{p(Y, Z \mid X, S; \beta, \eta^2, \sigma^2)}_{\mathcal{M}_{\text{FDA}}} = \underbrace{p(Y \mid X, S; \beta, \eta^2)}_{\mathcal{M}_{\text{fair}}} \underbrace{p(Z \mid Y; \sigma^2)}_{\mathcal{M}_{\text{faithful}}}. \tag{4}$$

Note, under the model $\mathcal{M}_{\text{faithful}}$, where $Z$ depends solely on $Y$, the conditional independence of $Z$ on $X$ and $S$ given $Y$ implies that $p(Z \mid Y, X, S) = p(Z \mid Y)$.

Followed by the joint model in equation 4, the synthetic values of $Y_i$, denoted as $\hat{Y}_i$ for $i \in [n]$, are then drawn from the corresponding predictive distribution as follows:

$$\hat{Y}_i \sim \ p\left(Y_i | Z_i, X_i, S_i; \hat{\beta}, \hat{\eta}^2, \hat{\sigma}^2\right)$$
$$\propto \underbrace{p(Y_i, Z_i \mid X_i, S_i; \hat{\beta}, \hat{\eta}^2, \hat{\sigma}^2)}_{\mathcal{M}_{\text{FDA}}} = \underbrace{p(Y_i \mid X_i, S_i; \hat{\beta}, \hat{\eta}^2)}_{\mathcal{M}_{\text{fair}}} \underbrace{p(Z_i \mid Y_i; \hat{\sigma}^2)}_{\mathcal{M}_{\text{faithful}}}, \tag{5}$$

where $\hat{\beta}, \hat{\eta}^2, \hat{\sigma}^2$ are the estimates of model parameters $\beta, \eta^2, \sigma^2$, respectively. Figure 2 presents a graphical representation of our data augmented joint modeling framework and fair synthetic data generation process.

With the intentional choice of Gaussian models for both $\mathcal{M}_{\text{fair}}$ and $\mathcal{M}_{\text{faithful}}$, it can be easily shown that the predictive distribution in equation 5 corresponds to the following Gaussian distribution (the proof is given in Appendix B.1):

$$\mathcal{N}\left(\frac{\frac{\hat{\sigma}^2}{M} f(X_i, S_i, \hat{\beta}) + \frac{\sum_{j=1}^M Z_{ij}}{M}\hat{\eta}^2}{\frac{\hat{\sigma}^2}{M} + \hat{\eta}^2}, \frac{\frac{\hat{\sigma}^2}{M}\hat{\eta}^2}{\frac{\hat{\sigma}^2}{M} + \hat{\eta}^2}\right). \tag{6}$$

In summary, given the estimates of the model parameters for both $\mathcal{M}_{\text{fair}}$ and $\mathcal{M}_{\text{faithful}}$, i.e., $\hat{\beta}, \hat{\eta}^2, \hat{\sigma}^2$, our synthetic dataset $\hat{\mathcal{D}} = \{\hat{Y}_i, X_i, S_i\}_{i=1}^n$ can be conveniently obtained by generating $\hat{Y}_i$, for $i \in [n]$ from the corresponding predictive distribution given in equation 6. Algorithm 1 summarizes the key steps using our FDA framework.

As shown in equation 6, it is clear that both $\mathcal{M}_{\text{fair}}$ and $\mathcal{M}_{\text{faithful}}$ contribute information to the synthetic values of $Y_i$'s. At one extreme, when $M = 0$ or/and $\sigma^2 = \infty$, the Gaussian distribution in equation 6 reduces to $\mathcal{N}\left(f(X_i, S_i, \hat{\beta}), \hat{\eta}^2\right)$; that is, the synthetic data $\hat{Y}_i$ will be generated based on the fair model $\mathcal{M}_{\text{fair}}$ alone, resulting in synthetic data $\hat{\mathcal{D}}$ that satisfies exact DP constraint. At the other extreme, when $M = \infty$ or/and $\sigma^2 = 0$, the Gaussian distribution in equation 6 degenerates to a point mass at $Y_i$ with $\bar{Z}_i = \frac{1}{M}\sum_{l=1}^M Z_{il} \overset{d}{=} Y_i$; that is, the information in $\mathcal{M}_{\text{faithful}}$ will completely override that of $\mathcal{M}_{\text{fair}}$, resulting in $\hat{Y}_i = Y_i$. Thus, when $M, \sigma^2 \in (0, \infty)$, an intermediate level of fairness and faithfulness to the original data will be achieved.

*Remark* 3.1 (Choosing tuning parameters in practice). To enhance the practical utility of our FDA framework, we further introduce a bias reduction factor, denoted as $\alpha = \frac{\eta^2}{\eta^2+\lambda}$ with $\lambda = \sigma^2/M$. This allows us to express the unfairness measure in the synthetic dataset, $\mathcal{UF}(\mathcal{P}_{\hat{\mathcal{D}}})$, in closed form

---

**Algorithm 1** FDA fair synthetic data generation algorithm

---

**Input:** original dataset $\mathcal{D} = \{X_i, Y_i, S_i\}_{i=1}^n$ and user specified $\lambda$.

**1.** Generate noisy copies $\{Z_i\}_{i=1}^n$ of $Y_i$ under the chosen faithful model $\mathcal{M}_{\text{faithful}}$.

**2.** Given the augmented data $\mathcal{D}_{\text{DA}} = \{\mathcal{D}, \{Z_i\}_{i=1}^n\}$,

**(2a).** Fit the joint model defined in equation 4 and obtain the parameter estimates $\hat{\eta}^2$, $\hat{\beta}$ and $\hat{\sigma}^2$;

**(2b).** Sample random draws $\hat{Y}_i$ from (6) for each $i \in [n]$.

**3.** Output fair synthetic datasets $\hat{\mathcal{D}} = \{\hat{Y}_i, X_i, S_i\}_{i=1}^n$.

---

as a function of this bias reduction factor. At the same time, we can quantify the Wasserstein-2 distance between the distribution of $Y_i$'s in the original dataset $\mathcal{D}$ and that of $\hat{Y}_i$'s in the synthetic dataset $\hat{\mathcal{D}}$ as a function of $\alpha$. By balancing the reduction of unfairness with the preservation of faithfulness to the original data, users can select an appropriate value of $\alpha$, which in turns can determine the values of the tuning parameters $M$ and $\sigma^2$ in the faithful model $\mathcal{M}_{\text{faithful}}$ given the relationship $\lambda = \sigma^2/M = \frac{(1-\alpha)\eta^2}{\eta^2}$. Further details are provided in Theorem 3.5, Theorem 3.6, and the subsequent discussions following these theorems.

*Remark* 3.2 (FDA beyond DP fairness). The FDA framework can be applied to generate fair synthetic data to satisfy the conditional statistical fairness (see Appendix A.4). In this case, one simply needs to choose $\mathcal{M}_{\text{fair}}$ such that it satisfies the conditional fairness criterion. For example, the constant mean model for $\mathcal{M}_{\text{fair}}$ mentioned earlier (i.e., CMM) naturally satisfies this fairness notion.

*Remark* 3.3 (FDA when combined with GAN-based fair models). We can let $\mathcal{M}_{\text{fair}}$ be a GAN-based fair data generator, e.g., DECAF. In this case, the generated synthetic data are guaranteed to achieve a higher level of fidelity to the original data compared to using DECAF alone (see Theorem 3.7).

*Remark* 3.4 (FDA framework when $Y_i$'s are categorical). In the case of categorical labels, a sample drawn from equation 6 is continuous. Then we need to map these continuous values back to discrete categories. One option is to round the continuous value to the nearest integer, or alternatively, one could define specific ranges for each category and map the continuous value to a category based on these predefined ranges. For example, when $Y \in \{0, 1\}$, one can use a threshold value $0.5$ as a cutoff to distinguish the two categories.

### 3.1 THEORETICAL ANALYSIS OF THE RESULTING SYNTHETIC DATA

Quantifying the fairness and faithfulness of synthetic data relative to the original data has been a longstanding challenge in previous fairness studies. While some prior works provide analytic bounds to characterize these properties, our work is the first to offer a closed-form solution, enabling an exact quantification of the trade-off between fairness and faithfulness. In this section, we proide a detailed discussion on how the synthetic data $\hat{\mathcal{D}}$ generated by our FDA framework achieves an $\alpha$-reduction in unfairness, where $\alpha$ is the bias reduction factor introduced in Remark 3.1, and how its faithfulness to the original data can be explicitly expressed as a function of $\alpha$.

**Theorem 3.5** (Unfairness reduction guarantee). *With the bias reduction factor $\alpha = \frac{\eta^2}{\eta^2 + \lambda}$, where $\eta^2$ is determined by $\mathcal{M}_{fair}$ and $\lambda = \sigma^2/M$, determined by the tuning parameters $\sigma^2$ and $M$ in $\mathcal{M}_{faithful}$, the distribution of the synthetic data $\hat{\mathcal{D}}$ generated by our FDA framework achieves the $\alpha$-reduction of unfairness as follows,*

$$\mathcal{UF}(\mathcal{P}_{\hat{\mathcal{D}}}) = \alpha \mathcal{UF}(\mathcal{P}_{\mathcal{D}}). \tag{7}$$

*where $\mathcal{P}_{\hat{\mathcal{D}}}$ and $\mathcal{P}_{\mathcal{D}}$ represent the distributions, from which the synthetic data $\hat{\mathcal{D}}$ and the original data $\mathcal{D}$ are sampled, respectively.*

By selecting the tuning parameters $\sigma^2$ and $M$ to achieve the desired value of $\alpha$, Theorem 3.5 guarantees the $\alpha$-reduction of unfairness on $\mathcal{P}_{\hat{\mathcal{D}}}$ for the resulting synthetic data $\hat{\mathcal{D}}$ (see proof in Appendix B.4). In practice, when $\eta$ is unknown, an estimate of $\eta$, denoted as $\hat{\eta}$, can be obtained by fitting the assumed $\mathcal{M}_{\text{fair}}$ first. Then, the values of $\sigma^2$ and $M$ can be chosen such that their ratio $\lambda = \hat{\eta}^2(1-\alpha)/\alpha$. Meanwhile, the following theorem quantifies the Wasserstein-2 distance between the conditional distribution of $\hat{Y}_i$'s in $\hat{\mathcal{D}}$ and that of the original $Y_i$'s in $\mathcal{D}$, when the synthetic data meets the desired fairness requirement characterized by any user-specified $\alpha$.

**Theorem 3.6** (Faithfulness quantification). *For the synthetic data $\hat{\mathcal{D}}$ such that its distribution $\mathcal{P}_{\hat{\mathcal{D}}}$ satisfies the $\alpha$-reduction of unfairness, the Wasserstein-2 distance between $\mu_{\hat{Y}|X,S}$ and $\mu_{Y|X,S}$ is given by*

$$W_2(\mu_{\hat{Y}|X,S}, \mu_{Y|X,S}) = \sqrt{(1-\alpha)^2 \mathbb{E}\left[(Y - f(X,S,\beta))^2 \,|X,S\right] + (1-\alpha^2)\,\eta^2}. \quad (8)$$

*where $\mu_{\hat{Y}|X,S}$ and $\mu_{Y|X,S}$ denote the conditional distribution of $\hat{Y}$'s given $X, S$ in $\hat{\mathcal{D}}$ and that of $Y_i$'s in $\mathcal{D}$, respectively. Particularly, when $M \to \infty$, and/or $\sigma^2 \to 0$, we have*

$$\hat{Y}|X,S \xrightarrow{p} Y|X,S, \text{ and consequently, } \hat{Y}|X,S \xrightarrow{d} Y|X,S, \quad (9)$$

*for any choice of $\mathcal{M}_{fair}$.*

Theorem 3.6 (see proof in Appendix B.2) quantifies the closeness of the generated synthetic values $\hat{Y}_i$'s to the original values $Y_i$'s with respect to the user-specified unfairness reduction factor $\alpha$. When $\lambda = 0$ (i.e., $M = \infty$ or $\sigma^2 = 0$), $\alpha = 1$ and $\hat{Y}$ is identical to $Y$; that is, $\hat{\mathcal{D}} = \mathcal{D}$ and $\mathcal{P}_{\hat{\mathcal{D}}}$ is as unfair as $\mathcal{P}_{\mathcal{D}}$. Conversely, when $\lambda = \infty$ (i.e., $M = 0$ and/or $\sigma^2 = \infty$), $\alpha = 0$ and the maximal discrepancy between $\hat{Y}$ and $Y$ is achieved, i.e., equation 8 attains it maximum $\sqrt{\mathbb{E}\left[(Y - f(X,S,\beta))^2 \,|X,S\right] + \eta^2}$, resulting in exact DP for $\mathcal{P}_{\hat{\mathcal{D}}}$ at the cost of faithfulness.

Theorem 3.6 and Theorem 3.5 together quantify the compromise one must make in order to meet the desired fairness requirement with respect to $\alpha$. Given a specific choice of $\mathcal{M}_{fair}$, a larger value of $\alpha$ results in fairer synthetic data $\hat{\mathcal{D}}$ but a greater discrepancy between the distributions of $\hat{Y}_i$'s and $Y_i$'s conditional on features. In practice, one can select a suitable $\alpha$ by balancing between the goals of reducing the unfairness and enhancing faithfulness. Importantly, achieving this goal can be facilitated by choosing appropriate values of the tuning parameters $\sigma^2$ and $M$ in $\mathcal{M}_{faithful}$. It is worth noting that the faithfulness in $\hat{\mathcal{D}}$ does depend on the predictor $f$ in the fair model $\mathcal{M}_{fair}$. Ideally, if one can find a fair predictor $f$ in $\mathcal{M}_{fair}$ that satisfies the DP constraint with minimal loss of faithfulness, then the generated synthetic data obtained using our FDA framework achieve the highest level of faithfulness to the original data while simultaneously meeting the desired fairness requirement with respect to $\alpha$.

Next, we demonstrate that the introduction of our faithful model $\mathcal{M}_{faithful}$ within our joint FDA framework ensures that the generated synthetic data are guaranteed to be more faithful to the original data compared to using the fair model $\mathcal{M}_{fair}$ alone.

**Theorem 3.7** (Faithfulness improvement guarantee). *For the synthetic data $\tilde{\mathcal{D}} := \{\tilde{Y}_i, X_i, S_i\}_{i=1}^n$ generated from the fair model $\mathcal{M}_{fair}$ alone (hence satisfying the exact DP constraint), the Wasserstein-2 distance between $\mu_{\tilde{Y}|X,S}$ and $\mu_{Y|X,S}$ is given by,*

$$W_2(\mu_{\tilde{Y}|X,S}, \mu_{Y|X,S}) = \sqrt{\mathbb{E}\left[(Y - f(X,S,\beta))^2 \,|X,S\right] + \eta^2}. \quad (10)$$

*Comparing equation 8 and equation 10, it is easy to see that,*

$$W_2(\mu_{\hat{Y}|X,S}, \mu_{Y|X,S}) < W_2(\mu_{\tilde{Y}|X,S}, \mu_{Y|X,S}) \quad (11)$$

*for any fixed $M > 0$ and $\sigma^2 > 0$, where $\mu_{\hat{Y}|X,S}$ denotes the conditional distribution of the synthetic $\hat{Y}_i$'s generated by our joint FAD framework using both $\mathcal{M}_{fair}$ and $\mathcal{M}_{faithful}$.*

Theorem 3.7 (see proof in Appendix B.3) proves that the synthetic values $\hat{Y}_i$'s obtained using FDA are closer to the original values $Y_i$'s than the synthetic values $\tilde{Y}_i$'s obtained from using $\mathcal{M}_{fair}$ alone. This finding provides provable guarantee that our joint FDA framework can be combined with any existing fair data generation model to further improve the faithfulness of the generated synthetic data. As we have discussed in the introduction, the generated synthetic dataset $\hat{\mathcal{D}}$ is used as the training dataset to train downstream models. For any trained downstream model $g(X, S)$ to predict $Y$, the model error could affect the downstream fairness. Theoretically, the upper bound of the fairness violation of the downstream model is given by the following proposition.

**Proposition 3.8.** *For any downstream model $g(X, S)$ to predict $Y$, if $W_2(\mu_{g(X,S)|s}, \mu_{\hat{Y}|s}) \leq \delta$ for all $s \in [K]$, we have*

$$\min_{\nu \in \mathcal{P}_p(\mathbb{R})} \sum_{s=1}^{K} \omega_s W_2(\mu_{g(X,S)|s}, \nu) \leq \alpha \mathcal{UF}(\mathcal{P}_\mathcal{D}) + \delta \,, \tag{12}$$

*for any given weights $(\omega_1, \cdots, \omega_K) \in \Delta^{K-1}$.*

Proposition 3.8 (see proof in Appendix B.5) establishes a uniform upper bound on fairness violations in the downstream models trained on the generated synthetic dataset. This upper bound can help users to select an appropriate $\alpha$ to ensure the desired level of downstream model fairness. Additionally, Proposition 3.8 suggests that the downstream model error, which is captured by $\delta$, can negatively affect the downstream model's fairness: a smaller error in the downstream model corresponds to improved fairness guarantees in the downstream models.

## 4 EXPERIMENTS

We demonstrate the novel features of our FDA framework in generating fair synthetic data and compare them with the baseline methods listed in Table 1 based on real data experiments. Further, we show that how our FDA framework can be combined with DECAF, a GAN-based fair data generator to improve the utility in downstream models trained on the resulting fair synthetic data.

All the baseline methods in Table 1 require intensive training and are extremely sensitive to the architecture and hyperparameters of the models. For example, different constructed causal graphs used by DECAF lead to varying utility in the resulting synthetic data. Therefore, to ensure a fair comparison, we run experiments on the UCI Adult dataset (Dua & Graff, 2020), which is the only dataset used by all the baseline methods. We also use the same model specifications as in the original code provided by the authors to maintain consistency in our comparison. The UCI Adult dataset contains over 65,000 samples with 11 attributes. It is known to exhibit bias between *gender* and *income* (Feldman et al., 2015b; Zhang et al., 2017a). Thus, we treat *gender* as the sensitive attribute and *income* as the binary target variable representing whether an adult's income is over $50K or not. Additional important details regarding the experimental setup, including the dataset split, the architecture of the downstream model, and its training process, are provided in Appendix C. Note that, we also conducted experiments on the COMPAS dataset using our FDA method. Due to space limitations, these results are included in Appendix D.

### 4.1 ACHIEVING TRADE-OFF BETWEEN FAIRNESS AND FAITHFULNESS IN DEBIASED SYNTHETIC DATA BY FDA

In this section, we show how our FDA framework facilitates the trade-off between absolute fairness ($\alpha = 0$) and perfect data faithfulness ($\alpha = 1$) by varying the bias reduction factor $\alpha$ within $(0, 1)$. Specifically, synthetic Adult datasets are generated using the baseline methods and our FDA under different values of $\alpha$. We repeat the experiments 10 times for each method. Figure 2 shows (1) the estimated Wasserstein-2 distance between the synthetic and original data distributions, denoted by $\hat{W}_2(\mu_{\hat{Y}}, \mu_Y)$, and (2) the estimated unfairness measure in the synthetic dataset, denoted by $\widehat{\mathcal{UF}}(\mathcal{P}_{\hat{\mathcal{D}}})$, obtained by each method, where the solid line represents the average of the 10 experiments and shadowed areas indicates variation[5]. As expected, when $\alpha \to 0$, $\widehat{\mathcal{UF}}(\mathcal{P}_{\hat{\mathcal{D}}}) \to 0$. This is when synthetic data achieves perfect fairness but the worst faithfulness to the original data, with $\hat{W}_2(\mu_{\hat{Y}}, \mu_Y)$ far away from 0. Conversely, when $\alpha \to 1$, $\hat{W}_2(\mu_{\hat{Y}}, \mu_Y) \to 0$. This is when the synthetic data distribution converges to the original data distribution, achieving perfect data faithfulness; in the meanwhile, the synthetic data achieves zero reduction of biases compared to the original data. None of the baseline methods offer this tuning mechanism, highlighting a unique advantage of our approach. It is also worth noting that, unlike other methods, our FDA shows very small variation that it is almost invisible on the plots, showing the stability of our FDA method.

---

[5]We observe similar patterns with the total variation-based unfairness measure detailed in the Appendix C.2, experiments on COMPAS dataset is given in Appendix D.

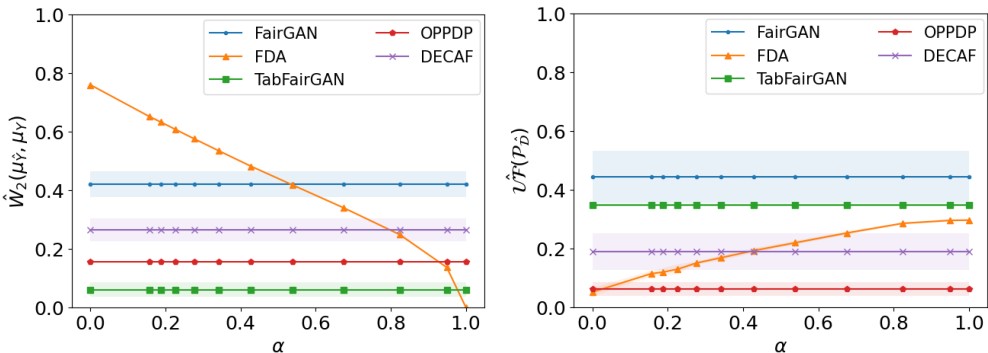

Figure 2: Faithfulness and fairness of the synthetic datasets by FairGAN, OPPDP, TabFairGAN, DECAF and FDA under varying values of $\alpha$: $\hat{W}_2(\mu_{\hat{Y}}, \mu_Y)$ ($\downarrow$ more faithful), $\widehat{\mathcal{UF}}(\mathcal{P}_{\hat{\mathcal{D}}})$ ($\downarrow$ more fair). The shadowed areas along each line represent the variations on 10 repetitions of experiments.

## 4.2 Improving data faithfulness and downstream utility using FDA

In this section, we conduct experiments on the Adult dataset to demonstrate how our FDA framework when combined with DECAF, can enhance the faithfulness of synthetic data to the original data and thereby enhances the performance of downstream models trained on the synthetic data while achieving an intermediate level of fairness.

Specifically, in our FDA framework, we let $\mathcal{M}_{\text{fair}}$ be the GAN-based model DECAF (van Breugel et al., 2021) (we term the corresponding model FDA-DECAF). To emphasize that, in our original FDA framework, $\mathcal{M}_{\text{fair}}$ is a constant mean model, namely CMM, we term the corresponding model FDA-CMM. We generate debiased synthetic data using DECAF, FDA-DECAF and FDA-CMM under different values of $\alpha$ and afterwards evaluate the utility and fairness of a downstream model trained on the resulting debiased synthetic datasets using these methods. Again, to ensure fair comparisons, we follow van Breugel et al. (2021) and focus on the same downstream MLP model. When evaluating the utility and fairness of the downstream model trained on the debiased synthetic data, we focus on the same metrics: (1) utility: we evaluate the predictive performance of the model using accuracy, precision, recall, and AUROC; (2) fairness: we assess the fairness of the downstream model trained on the synthetic data using the total variation distance based measure $|\mathbb{P}(\hat{Y}|S = 1) - \mathbb{P}(\hat{Y}|S = 0)|$.

Figure 3 shows the results[6] of our experiments, repeated 10 times. As expected, FDA-DECAF leads to consistently better utility in the downstream prediction for any $\alpha \in (0, 1)$ when compared with DECAF. This finding is supported by our theoretical result given in Theorem 3.7. By introducing a joint modeling approach (including $\mathcal{M}_{\text{fair}}$ and $\mathcal{M}_{\text{faithful}}$), coupled with the tuning mechanism to allow for an intermediate level of fairness, our FDA framework enables the resulting synthetic data to maintain a certain level of faithfulness to the original data, thereby enhancing the downstream prediction performance. When $\alpha \to 1$, our FDA framework allows the utility of the downstream model trained on the resulting synthetic data to fully recover the utility using the original data. In contrast, DECAF enforces exact fairness in the synthetic data, which can lead to significant loss of utility in the downstream model as shown in Figure 3.

It is also worth noting that, despite the greater efforts required to implement the FDA-DECAF method, which involves constructing causal graphs and intensive model tuning, FDA-DECAF does not perform better than our FDA-CMM, which only involves sampling from Gaussian distributions. When achieving similar utility-fairness trade-off, our FDA-CMM method offers significant advantages in terms of computational efficiency, ease of implementation, stability and interpretability.

## 5 Discussion

In this paper, we introduce the FDA framework to generate debiased synthetic data, aiming to achieve intermediate levels of fairness and faithfulness as controlled by a user-specified unfairness

---

[6]See Appendix C.3 for more implementation details.

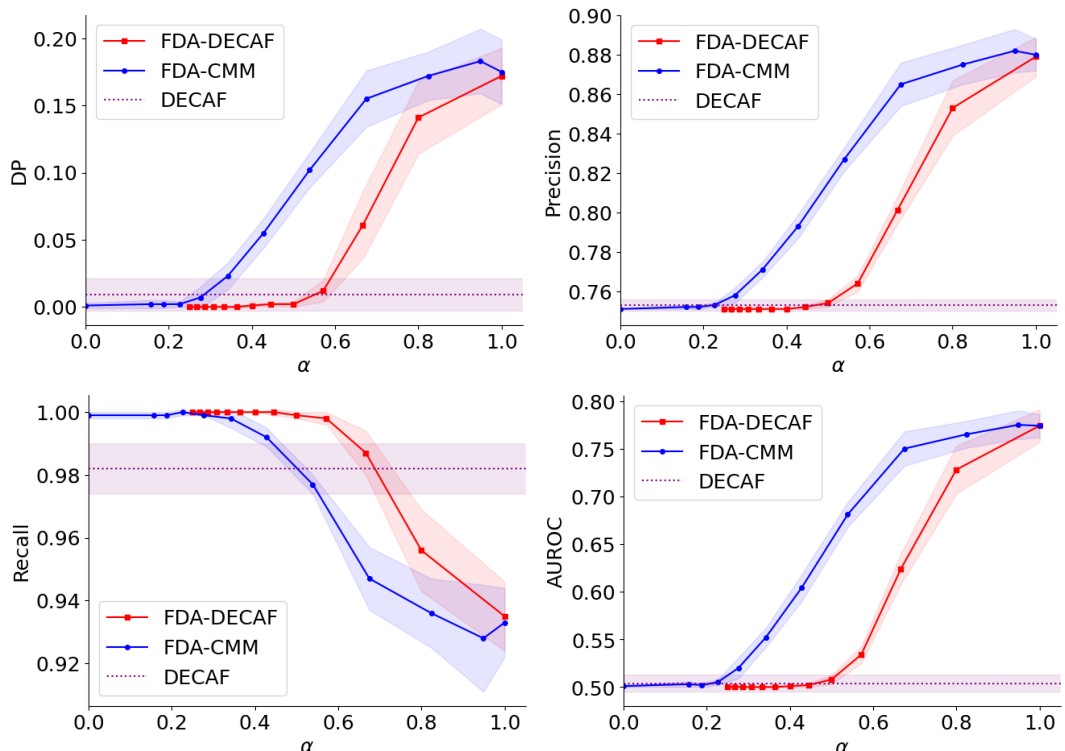

Figure 3: Utility and fairness of the downstream models trained on the synthetic datasets obtained by DECAF, FDA-DECAF and FDA-CMM under varying values of $\alpha$: DP↓ (top left), Precision↑ (top right), Recall↑ (bottom left), AUROC↑ (bottom right). The shadowed areas along each line represent the variations across 10 repeated experiments.

reduction factor ($\alpha$). This is the first work to offer a provable trade-off characterization between these two competing objectives. Our experimental results demonstrate the effectiveness of FDA in achieving this trade-off, and validate our theoretical guarantees: achieving perfect data faithfulness when setting $\alpha = 1$, which is an important feature not present in other methods, and absolute fairness when setting $\alpha = 0$. Moreover, our FDA framework improves the faithfulness of synthetic data when integrated with DECAF, a GAN-based fair data generator, as both proven theoretically and demonstrated empirically in our experiments. This enhancement in faithfulness, in turn, boosts the utility of downstream models trained on the resulting synthetic data.

**Social implications.** Setting $\alpha$ at different values allows users to balance the trade-off between absolute fairness and perfect data faithfulness, adapting to the specific needs and priorities of various applications. This capability is crucial, as it enables decision-makers to make informed choices about this trade-off without having to sacrifice one aspect for the other. Our FDA framework represents a significant advancement in synthetic data generation, offering a transparent and robust approach that involves only sampling from Gaussian distributions shown in Equation equation 6 and relies on no assumptions. Such simplicity contrasts sharply with the labor-intensive and frequently unstable training processes of complex black-box methods. These features of our FDA framework make it more accessible, allowing a broader range of practitioners to adopt it without requiring specialized training. This could potentially transform how data is managed to ensure equity and fairness.

**Future work.** The current FDA framework addresses the generation of fair synthetic data for one-dimensional labels. Extending this framework to multi-dimensional labels, including mixed labels of continuous and categorical types, is computationally straightforward; however, the challenge lies in the theoretical analysis required to establish the relationship between fairness and faithfulness in this context. Addressing this challenge will be the focus of our future work.

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

## A  DEFINITIONS AND NOTIONS

### A.1  P-WASSERSTEIN DISTANCE

**Definition A.1** (p-Wasserstein distance). Let $\mu$ and $\nu$ be two probability measures on $\mathbb{R}$. For $p \geq 1$, the p-Wasserstein distance between $\mu$ and $\nu$ is defined as

$$
W_p(\mu, \nu) = \left[ \inf_{\gamma \in \Gamma(\mu, \nu)} \int |x - y|^p d\gamma(x, y) \right]^{\frac{1}{p}},
$$

where $\Gamma(\mu, \nu)$ is the set of joint probability measures on $\mathbb{R} \times \mathbb{R}$ with marginals are $\mu$ and $\nu$. Namely, for all measurable sets $A, B \subseteq \mathbb{R}$, it holds that $\gamma(A \times \mathbb{R}) = \mu(A)$ and $\gamma(\mathbb{R} \times B) = \nu(B)$.

### A.2  EVALUATION OF THE UNFAIRNESS MEASURE $\mathcal{UF}(\mathcal{P}_\mathcal{D})$

The closed-form computation of $\mathcal{UF}(\mathcal{P}_\mathcal{D})$ for any $\mathcal{P}_\mathcal{D}$ is not easy due to the complex computation of Wasserstein-2 distance in equation 1. In the following example, we present one example when $\mathcal{UF}(\mathcal{P}_\mathcal{D})$ can be explicitly computed.

Consider the case when $\mu_{Y|s} = \mathcal{N}(b_s, \sigma_s^2)$ is a normal distribution, then it is known the minimizer (Agueh & Carlier, 2011) of equation 1 is $\nu = \mathcal{N}\left( \sum_{s=1}^{K} \omega_s b_s, \left( \sum_{s=1}^{K} \omega_s \sigma_s \right)^2 \right)$. Therefore, one can compute $\mathcal{UF}(\mathcal{P}_\mathcal{D})$ (Dowson & Landau, 1982) as

$$
\mathcal{UF}(\mathcal{P}_\mathcal{D}) = \sum_{s=1}^{K} \omega_s \left[ \left( b_s - \sum_{s=1}^{K} \omega_s b_s \right)^2 + \left( \sigma_s - \sum_{s=1}^{K} \omega_s \sigma_s \right)^2 \right]^{\frac{1}{2}}.
$$

In practice, the unfairness measure can be estimated by replacing the Wasserstein-2 distance in equation 1 with its estimator. There are many well studied estimators of Wasserstein-2 distance, for example, the plug-in estimator (Sommerfeld & Munk, 2018; Tameling et al., 2019; Dvurechensky et al., 2018), the estimation based on Sinkhorn divergence (Chizat et al., 2020).

### A.3  INTERPRETATION OF THE WEIGHTS IN EQUATION 1

The unfairness measure provides flexibility in choosing different weights $(\omega_1, \cdots, \omega_K)$ to accommodate various purposes, particularly when there are majority and/or minority groups with respect to the sensitive attribute $S$. For example, the majority group (with respect to $s_{\text{majority}}$) can be identified when $\mathbb{P}(S = s_{\text{majority}}) \gg \mathbb{P}(S = s)$ for all $s \in [K] \setminus \{s_{\text{majority}}\}$.

In general, for any $s \in [K]$, the larger the $\omega_s$, the closer the $\mu_{Y|X}$ is to the optimal $\nu$ (the minimizer in equation 1). This leads to some natural choices of $(\omega_1, \cdots, \omega_K)$, including $\omega_s = \mathbb{P}(S = s)$ and $\omega_s \propto 1/\mathbb{P}(S = s)$. When using the former, the optimal $\nu$ will be closer to the conditional distribution of $Y$ for the majority group; when using the latter, the optimal $\nu$ will be closer to the conditional distribution of $Y$ for the minority group. Alternatively, one could use equal weights by letting $\omega_s = \frac{1}{K}$, when all groups are similar in size.

### A.4  FAIRNESS NOTIONS

**Definition A.2** (Conditional Fairness). Let $X = (\tilde{X}, F)$, $\mathcal{P}_\mathcal{D}$ is said to satisfy conditional fairness with respect to , if $(Y|S = s_1, F = f) \overset{d}{=} (Y|S = s_2, F = f)$, for any $s_1, s_2 \in [K]$. That is to say, $\mathbb{P}(Y \in T|S = s_1, F = f) = \mathbb{P}(Y \in T|S = s_2, F = f)$ for any $T \subseteq \mathbb{R}$.

## B  PROOFS FOR MAIN RESULTS IN SECTION 3

The detailed proofs of the results in the main paper are included in this appendix.

## B.1 PROOF OF EQUATION EQUATION 6

*Proof.* As discussed in the main paper, given the estimates of the model parameters $(\hat{\beta}, \hat{\eta}^2, \hat{\sigma}^2)$, we draw the synthetic $\hat{Y}_i$ from the following predictive distribution defined under $\mathcal{M}_{\text{FDA}}$:

$$p(Y_i|Z_i, X_i, S_i; \hat{\beta}, \hat{\eta}^2, \hat{\sigma}^2) \propto \underbrace{p(Y_i|X_i, S_i; \hat{\eta}^2, \hat{\beta})}_{\mathcal{M}_{\text{fair}}} \underbrace{p(Z_i|Y_i; \hat{\sigma}^2)}_{\mathcal{M}_{\text{faithful}}}. \tag{13}$$

Under the fair model $\mathcal{M}_{\text{fair}}$ defined in equation 2, we have

$$p(Y_i|X_i, S_i; \hat{\eta}^2, \hat{\beta}) = \mathcal{N}\left(f(X_i, S_i, \hat{\beta}), \hat{\eta}^2\right),$$

and under the faithful model $\mathcal{M}_{\text{fair}}$ defined in equation 3, we have

$$p(Z_i|Y_i; \hat{\sigma}^2) = \prod_{m=1}^{M} \mathcal{N}\left(Y_i, \hat{\sigma}^2\right).$$

That is,

$$p(Y_i|X_i, S_i; \hat{\beta}, \hat{\eta}^2) \propto \exp\left\{-\frac{\left(Y_i - f\left(X_i, S_i, \hat{\beta}\right)\right)^2}{2\hat{\eta}^2}\right\}$$

and

$$p(Z_i|Y_i; \hat{\sigma}^2) \propto \prod_{m=1}^{M} \exp\left\{-\frac{(Y_i - Z_{im})^2}{2\hat{\sigma}^2}\right\}$$

Then, realizing that

$$\exp\left\{-\frac{\left(Y_i - f\left(X_i, S_i, \hat{\beta}\right)\right)^2}{2\hat{\eta}^2}\right\} \prod_{m=1}^{M} \exp\left\{-\frac{(Y_i - Z_{im})^2}{2\hat{\sigma}^2}\right\}$$

$$\propto \exp\left\{-\frac{1}{2}\left(\frac{\frac{\hat{\sigma}^2}{M}\hat{\eta}^2}{\frac{\hat{\sigma}^2}{M} + \hat{\eta}^2}\right)^{-1}\left(\hat{Y}_i - \frac{\frac{\hat{\sigma}^2}{M}f\left(X_i, S_i, \hat{\beta}\right) + \frac{\sum_{m=1}^{M} Z_{im}}{M}\hat{\eta}^2}{\frac{\hat{\sigma}^2}{M} + \hat{\eta}^2}\right)^2\right\},$$

which corresponds to the kernel of the Gaussian distribution as defined in equation 6:

$$\mathcal{N}\left(\frac{\frac{\hat{\sigma}^2}{M}f\left(X_i, S_i, \hat{\beta}\right) + \frac{\sum_{m=1}^{M} Z_{im}}{M}\hat{\eta}^2}{\frac{\hat{\sigma}^2}{M} + \hat{\eta}^2}, \frac{\frac{\hat{\sigma}^2}{M}\hat{\eta}^2}{\frac{\hat{\sigma}^2}{M} + \hat{\eta}^2}\right).$$

This concludes the proof of equation 6. $\qquad\square$

For clarity and ease of reading, we present the proof of Theorem 3.6 before Theorem 3.5.

## B.2 PROOF OF THEOREM 3.6

*Proof.* To determine the Wasserstein distance between $\mu_{\hat{Y}|X,S}$ and $\mu_{Y|X,S}$, it is convenient to rewrite the generating model as

$$\hat{Y} = \frac{\frac{\sigma^2}{M}f(X, S, \beta)}{\frac{\sigma^2}{M} + \eta^2} + \frac{\frac{\sum_{j=1}^{M} Z_j}{M}\eta^2}{\frac{\sigma^2}{M} + \eta^2} + \sqrt{\frac{\frac{\sigma^2}{M}\eta^2}{\frac{\sigma^2}{M} + \eta^2}}N_1,$$

$$Z_j = Y + \sigma N_2, \quad \text{for } j = 1, \cdots, M,$$

where $N_1$ and $N_2$ are independent standard normal random variables that is independent of $Y$, $\{Z_j\}_{j=1}^{M}$ are $M$ noisy copies of $Y$ (we omit the individual index $i$ for simplicity). Thus,

$$\hat{Y} = \frac{\frac{\sigma^2}{M}f(X, S, \beta)}{\frac{\sigma^2}{M} + \eta^2} + \frac{Y\eta^2}{\frac{\sigma^2}{M} + \eta^2} + \sqrt{\frac{\frac{\sigma^2}{M}\eta^2}{\frac{\sigma^2}{M} + \eta^2}}N_1 + \frac{\eta^2}{\frac{\sigma^2}{M} + \eta^2}\sqrt{\frac{\sigma^2}{M}}N_2.$$

Followed by $\hat{Y} - Y = \frac{\frac{\sigma^2}{M} f(X,S,\beta)}{\frac{\sigma^2}{M} + \eta^2} - \frac{Y \frac{\sigma^2}{M}}{\frac{\sigma^2}{M} + \eta^2} + \sqrt{\frac{\frac{\sigma^2}{M} \eta^2}{\frac{\sigma^2}{M} + \eta^2}} N_1 + \frac{\eta^2}{\frac{\sigma^2}{M} + \eta^2} \sqrt{\frac{\sigma^2}{M}} N_2$, one has

$$W_2^2(\mu_{\hat{Y}|X,S}, \mu_{Y|X,S}) = \inf_{\gamma \in \Gamma(\mu_{\hat{Y}|X,S}, \mu_{Y|X,S})} \int (\hat{y} - y)^2 d\gamma(\hat{y}, y)$$

$$= \mathbb{E}\left[ \left( \frac{\frac{\sigma^2}{M} f(X,S,\beta)}{\frac{\sigma^2}{M} + \eta^2} - \frac{Y \frac{\sigma^2}{M}}{\frac{\sigma^2}{M} + \eta^2} + \sqrt{\frac{\frac{\sigma^2}{M} \eta^2}{\frac{\sigma^2}{M} + \eta^2}} N_1 + \frac{\eta^2}{\frac{\sigma^2}{M} + \eta^2} \sqrt{\frac{\sigma^2}{M}} N_2 \right)^2 \Bigg| X, S \right]$$

$$= \left( \frac{\sigma^2}{\sigma^2 + M\eta^2} \right)^2 \mathbb{E}\left[ (Y - f(X,S,\beta))^2 | X, S \right] + \frac{\sigma^2 \eta^2}{\sigma^2 + M\eta^2} + \frac{M\eta^4 \sigma^2}{(\sigma^2 + M\eta^2)^2},$$

where the last equation is a direct computation by taking the expectation of the squared form. The proof is competed by the relationships $\frac{\lambda \eta^2}{\lambda + \eta^2} + \frac{\eta^4 \lambda}{(\lambda + \eta^2)^2} = (1 - \alpha)\eta^2 + \alpha(1 - \alpha)\eta^2 = (1 - \alpha^2)\eta^2$.

To see the asymptotic result when $M \to \infty$ or/and $\sigma^2 \to 0$, we shall check the moments of $\hat{Y} - Y$.

$$\mathbb{E}\left[ (\hat{Y} - Y)^2 | X, S \right] \tag{14}$$

$$= \mathbb{E}\left[ \left( \frac{\frac{\sigma^2}{M} f(X,S,\beta)}{\frac{\sigma^2}{M} + \eta^2} - \frac{Y \frac{\sigma^2}{M}}{\frac{\sigma^2}{M} + \eta^2} + \sqrt{\frac{\frac{\sigma^2}{M} \eta^2}{\frac{\sigma^2}{M} + \eta^2}} N_1 + \frac{\eta^2}{\frac{\sigma^2}{M} + \eta^2} \sqrt{\frac{\sigma^2}{M}} N_2 \right)^2 \Bigg| X, S \right]$$

$$= \left( \frac{\sigma^2}{\sigma^2 + M\eta^2} \right)^2 \mathbb{E}\left[ (Y - f(X,S,\beta))^2 | X, S \right] + \frac{\sigma^2 \eta^2}{\sigma^2 + M\eta^2} + \frac{M\eta^4 \sigma^2}{(\sigma^2 + M\eta^2)^2}. \tag{15}$$

It is trivial to see from equation 15 converges to 0 when $M \to \infty$ or/and $\sigma^2 \to 0$.

By Markov's inequality, for any $\epsilon > 0$,

$$\mathbb{P}(|\hat{Y} - Y| > \epsilon | X, S) \leq \frac{\mathbb{E}\left[ (\hat{Y} - Y)^2 | X, S \right]}{\epsilon^2}, \tag{16}$$

where the right hand side converges to 0 when $M \to \infty$ or/and $\sigma^2 \to 0$. Therefore, we have $\hat{Y}|X, S \to Y|X, S$ in probability.

$\square$

### B.3 PROOF OF THEOREM 3.7

The proof of Theorem 3.7 follows directly from the result in Theorem 3.6. The detail is given as follows.

*Proof.* If $\tilde{\mathcal{D}}$ is only generated from the fair model $\mathcal{M}_{\text{fair}}$, we have

$$\tilde{Y} - Y = f(X,S,\beta) - Y + \eta N_1,$$

where $N_1$ is a standard normal random variable that is independent of $Y$. Thus,

$$W_2^2(\mu_{\tilde{Y}|X,S}, \mu_{Y|X,S}) = \inf_{\gamma \in \Gamma(\mu_{\tilde{Y}}, \mu_Y)} \int (\tilde{y} - y)^2 d\gamma(\tilde{y}, y)$$

$$= \mathbb{E}\left[ (f(X,S,\beta) - Y + \eta N_1)^2 | X, S \right]$$

$$= \mathbb{E}\left[ (f(X,S,\beta) - Y)^2 | X, S \right] + \eta^2.$$

To obtain the inequality $W_2^2(\mu_{\hat{Y}|X,S}, \mu_{Y|X,S}) \leq W_2^2(\mu_{\tilde{Y}|X,S}, \mu_{Y|X,S})$, one can use the following results.

For any $\sigma^2$ and $M > 0$, we always have $\left(\frac{\sigma^2}{\sigma^2 + M\eta^2}\right)^2 \leq 1$, where the quality holds only when $M \to 0$ or/and $\sigma^2 \to \infty$.

On the other hand, it is trivial to obtain the following inequality.

$$\frac{\sigma^2\eta^2}{\sigma^2 + M\eta^2} + \frac{M\eta^4\sigma^2}{(\sigma^2 + M\eta^2)^2} = \eta^2\left(1 - \frac{M^2\eta^4}{(\sigma^2 + M\eta^2)^2}\right) \leq \eta^2\,,$$

where the equality of the last inequality holds only when $M \to 0$ or/and $\sigma^2 \to \infty$.

Thus, the inequality in Remark 3.7 is obtained immediately. $\qquad\square$

### B.4 PROOF OF THEOREM 3.5

*Proof.* First, due to the scaling law of Wasserstein-2 distance and its corresponding barycenter (Santambrogio, 2015; Panaretos & Zemel, 2019; Chzhen & Schreuder, 2022; Villani, 2021), we have

$$\min_{\nu \in \mathcal{P}_2(\mathbb{R})} \sum_{s=1}^{K} \omega_s W_p(\mu_{cY|s}, \nu) = c \min_{\nu \in \mathcal{P}_2(\mathbb{R})} \sum_{s=1}^{K} \omega_s W_p(\mu_{Y|s}, \nu)\,, \tag{17}$$

for any $c \geq 0$.

Secondly, by the translation invariant property of Wasserstein-2 distance (Santambrogio, 2015; Panaretos & Zemel, 2019; Villani, 2021), one has

$$W_2(\mu_{Y+Z+a}, \mu_{X+Z+a}) = W_2(\mu_Y, \mu_X)\,,$$

for any constant $a$ and random variable $Z$ that is independent of $Y$ and $X$. Thus,

$$\min_{\nu \in \mathcal{P}_2(\mathbb{R})} \sum_{s=1}^{K} \omega_s W_2(\mu_{Y+Z+a|s}, \nu) = \min_{\nu \in \mathcal{P}_2(\mathbb{R})} \sum_{s=1}^{K} \omega_s W_2(\mu_{Y|s}, \nu)\,. \tag{18}$$

Based on the generating model of $\hat{Y}$, we have

$$\hat{Y} = \frac{\frac{\sigma^2}{M}f(X, S, \beta)}{\frac{\sigma^2}{M} + \eta^2} + \frac{Y\eta^2}{\frac{\sigma^2}{M} + \eta^2} + \sqrt{\frac{\frac{\sigma^2}{M}\eta^2}{\frac{\sigma^2}{M} + \eta^2}}N_1 + \frac{\eta^2}{\frac{\sigma^2}{M} + \eta^2}\sqrt{\frac{\sigma^2}{M}}N_2\,,$$

where $N_1$ and $N_2$ are independent standard normal random variables that are independent of $Y$. A direct application of equation 17 and equation 18 implies $\mathcal{UF}(\mathcal{P}_{\hat{D}}) = \frac{\eta^2}{\frac{\sigma^2}{M} + \eta^2}\mathcal{UF}(\mathcal{P}_D) = \alpha\mathcal{UF}(\mathcal{P}_D)$ for any given weights $(\omega_1, \cdots, \omega_K) \in \Delta^{K-1}$. $\qquad\square$

### B.5 PROOF OF PROPOSITION 3.8

*Proof.* Assume $\nu_0 = \arg\min_{\nu \in \mathcal{P}_2(\mathbb{R})} \sum_{s=1}^{K} \omega_s W_2(\mu_{\hat{Y}|s}, \nu)$, by Theorem 3.5 we have

$$\sum_{s=1}^{K} \omega_s W_2(\mu_{\hat{Y}|s}, \nu_0) = \alpha\mathcal{UF}(\mathcal{P}_D)\,.$$

By triangle inequality, we have

$$\sum_{s=1}^{K} \omega_s W_2(\mu_{g(X,S)|s}, \nu_0) \leq \sum_{s=1}^{K} \omega_s W_2(\mu_{\hat{Y}|s}, \nu_0) + \sum_{s=1}^{K} \omega_s W_2(\mu_{g(X,S)|s}, \mu_{\hat{Y}|s}) \tag{19}$$

$$\leq \alpha\mathcal{UF}(\mathcal{P}_D) + \delta\,, \tag{20}$$

for any given $(\omega_1, \cdots, \omega_K)$. It follows that

$$\min_{\nu \in \mathcal{P}_2(\mathbb{R})} \sum_{s=1}^{K} \omega_s W_2(\mu_{g(X,S)|s}, \nu) \leq \sum_{s=1}^{K} \omega_s W_2(\mu_{g(X,S)|s}, \nu_0) \leq \alpha\mathcal{UF}(\mathcal{P}_D) + \delta\,.$$

$\qquad\square$

# C ADDITIONAL EXPERIMENTS ON UCI ADULT DATASET

## C.1 COMPUTATION DETAILS OF OUR EXPERIMENTS

**Data split:** The UCI Adult dataset is randomly split into two sets: a training set with $63,000$ data points and a testing set with data size $2,000$ data points.

**Model training:** The model parameter $\beta$ depends on the specific choice of $\mathcal{M}_{\text{fair}}$, for example, $\hat{\beta} = \bar{Y}$ if $\mathcal{M}_{\text{fair}}$ is CMM. Here, we estimate the parameters using their sample versions. That is, for the CMM model, $\hat{\beta} = \bar{Y}$ and $\hat{\eta} = \frac{1}{n-1}\sum_{i=1}^{n}(Y_i - \bar{Y})^2$. Additionally, since the hyperparameter $\sigma^2$ is tuned by the user, its estimation is not necessary and its true value is used. The average time spent for the training process to obtain one synthetic dataset is around 1.3 seconds on the Adult dataset with $n = 63,000$ in the training set. The simulations were performed using Python 3.8.8 on a PC with a 12th Gen intel Core i5-12600K CPU with 32 GB of RAM running Windows 11.

**Downstream model:** To evaluate the fairness and utility of downstream models, we train Multi-layer Perceptron (MLP) models on the generated synthetic datasets based on different fair data generation models. For instance, MLP models are trained using `MLPclassifier` function from `sklearn` module with all default parameters.

## C.2 ADDITIONAL FAIRNESS EVALUATION ON SYNTHETIC DATA BY FDA

In this section, we run experiments on the UCI Adult dataset $\mathcal{D}$. Synthetic dataset $\hat{\mathcal{D}}$ is generated under FDA framework with different choices of $\alpha$. In addition to the unfairness measure $\mathcal{UF}(\hat{\mathcal{D}})$ in Definition 2.2, we also assess the commonly used unfairness measure $|\mathbb{P}(\hat{Y} = 1|S = 1) - \mathbb{P}(\hat{Y} = 1|S = 0)|$ with respect to the bias reduction parameter $\alpha$ and illustrate in Figure 4. Still, the result of FDA is compared with those of DECAF (van Breugel et al., 2021), FairGAN (Xu et al., 2018), OPPDP(Calmon et al., 2017), TabFairGAN(Rajabi & Garibay, 2022). Each experiment is repeated 10 times, the average performances are reported in solid lines with shadowed variation areas in Figure 4. It is clear that, FDA still shows a clear tuning mechanism on faithfulness and fairness with respect to the bias reduction parameter $\alpha$, and the tuning trend is the same as using the unfairness measurement $\mathcal{UF}(\mathcal{P}_{\hat{\mathcal{D}}})$. This further emphasises FDA does facilitate the trade-off between absolute fairness and perfect data faithfulness by varying the bias reduction factor $\alpha$.

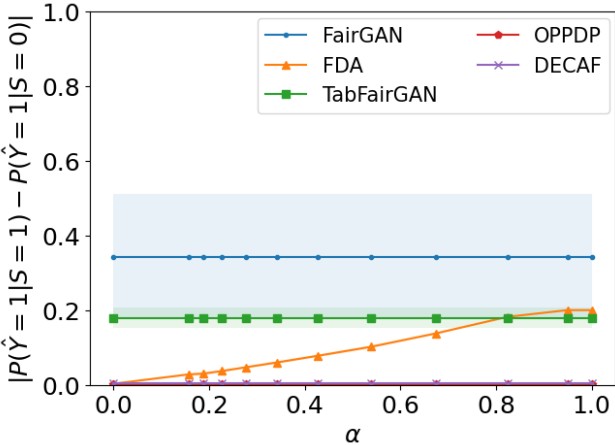

Figure 4: Fairness of the generated synthetic dataset by FDA, FairGAN, OPPDP, DECAF with respect to $\alpha$: $|\mathbb{P}(\hat{Y} = 1|S = 1) - \mathbb{P}(\hat{Y} = 1|S = 0)|$ ($\downarrow$ more fair). The shadowed areas along each line represent the variations on 10 repeat ions of experiments.

Table 2: Data utility and fairness of the downstream MLP model trained on generated synthetic data by FDA-DECAF model with different tuning ratio $\lambda$.

| FDA-DECAF | DATA UTILITY | | | FAIRNESS | |
|---|---|---|---|---|---|
| $\mathcal{M}_{\text{FAIR}}$ : DECAF | PRECISION ↑ | RECALL↑ | AUROC↑ | DP↓ | FTU↓ |
| ORIGINAL DATASET | $0.879 \pm 0.012$ | $0.933 \pm 0.012$ | $0.773 \pm 0.021$ | $0.182 \pm 0.019$ | $0.028 \pm 0.013$ |
| $\lambda = 0.25$ | $0.846 \pm 0.012$ | $0.962 \pm 0.001$ | $0.717 \pm 0.023$ | $0.142 \pm 0.023$ | $0.050 \pm 0.024$ |
| $\lambda = 0.5$ | $0.846 \pm 0.001$ | $0.958 \pm 0.011$ | $0.716 \pm 0.019$ | $0.141 \pm 0.025$ | $0.065 \pm 0.028$ |
| $\lambda = 1$ | $0.800 \pm 0.006$ | $0.989 \pm 0.005$ | $0.622 \pm 0.015$ | $0.071 \pm 0.017$ | $0.028 \pm 0.018$ |
| $\lambda = 2$ | $0.754 \pm 0.001$ | $1.000 \pm 0.000$ | $0.507 \pm 0.002$ | $0.002 \pm 0.002$ | $0.003 \pm 0.002$ |
| $\lambda = 4$ | $0.751 \pm 0.000$ | $1.000 \pm 0.000$ | $0.500 \pm 0.000$ | $0.000 \pm 0.001$ | $0.000 \pm 0.000$ |
| DECAF | $0.753 \pm 0.000$ | $0.989 \pm 0.000$ | $0.505 \pm 0.000$ | $0.006 \pm 0.000$ | $0.006 \pm 0.000$ |

Table 3: Data utility and fairness of the downstream MLP model trained on generated synthetic data by FDA-CMM model with different tuning ratio $\lambda$.

| FDA-CMM | DATA UTILITY | | | FAIRNESS | |
|---|---|---|---|---|---|
| $\mathcal{M}_{\text{FAIR}}$ : CMM | PRECISION ↑ | RECALL↑ | AUROC↑ | DP↓ | FTU↓ |
| ORIGINAL DATASET | $0.877 \pm 0.009$ | $0.934 \pm 0.009$ | $0.768 \pm 0.016$ | $0.169 \pm 0.022$ | $0.031 \pm 0.026$ |
| $\lambda = 0.25$ | $0.931 \pm 0.014$ | $0.781 \pm 0.037$ | $0.803 \pm 0.008$ | $0.287 \pm 0.031$ | $0.079 \pm 0.044$ |
| $\lambda = 0.5$ | $0.861 \pm 0.007$ | $0.928 \pm 0.019$ | $0.738 \pm 0.009$ | $0.155 \pm 0.041$ | $0.079 \pm 0.039$ |
| $\lambda = 1$ | $0.751 \pm 0.000$ | $1.000 \pm 0.000$ | $0.501 \pm 0.001$ | $0.001 \pm 0.001$ | $0.000 \pm 0.001$ |
| $\lambda = 2$ | $0.751 \pm 0.000$ | $1.000 \pm 0.000$ | $0.500 \pm 0.000$ | $0.000 \pm 0.000$ | $0.000 \pm 0.000$ |
| $\lambda = 4$ | $0.751 \pm 0.000$ | $1.000 \pm 0.000$ | $0.500 \pm 0.000$ | $0.000 \pm 0.000$ | $0.000 \pm 0.000$ |

### C.3 IMPROVING DATA FAITHFULNESS AND DOWNSTREAM UTILITY USING FDA BASED ON TUNING RATIO $\lambda$

We generate synthetic fair dataset by FDA-DECAF model and by FDA with different choices of $\lambda$, the evaluation of downstream faithfulness and fairness are presented and compared with DECAF in Table 2 and 3, respectively. As expected, It clearly shows the downstream faithfulness is decreasing when $\lambda$ is increasing, while the fairness is increasing simultaneously. This scenario coincides with the theoretical findings in Section 3.1. In addition to DP fairness, we also evaluate a different fairness notion: Fairness Through Unawareness (FTU), which is the difference between the predicted variables of a downstream classifier for setting $S = 1$ and $S = 0$, respectively, while giving the same feature. FTU is evaluated by the metric as $|\mathbb{P}_{S=1}(\hat{Y}|X) - \mathbb{P}_{S=0}(\hat{Y}|X)|$.

Comparing results in Table 2 for FDA-DECAF and Table 3 for FDA-CMM, it is interesting to see when the tuning ratio $\lambda$ is large (i.e., $\lambda \geq 2$) the data utility and fairness performances for FDA-DECAF and FDA-CMM coincide. That is to say, when high level of fairness is required, one can either use FDA-DECAF or FDA-CMM. However, it is clear that, not like FDA-DECAF (prior causal relationships knowledge is required), FDA-CMM is a very simple model with no prior knowledge requirement. FDA-CMM is very recommended due to its computation simplicity and less assumptions requirement. This scenario coincides the interpretation in Section 4.2 and its reason is when $\lambda$ is large, the information from $\mathcal{M}_{\text{faithful}}$ will significantly override the information from $\mathcal{M}_{\text{fair}}$, leading high fairness level of the synthetic dataset from FDA joint model.

## D ADDITIONAL EXPERIMENTS ON COMPAS DATASET

The proposed FDA framework is very general not only on the capability of using different fair model $\mathcal{M}_{\text{fair}}$, but also on the stability of its performance on various real data. To show the generalization of using FDA framework, we run experiments on COMPAS data (Angwin et al., 2016), which is a dataset contains information about defendants from Broward County, and contains attributes about defendants such as their ethnicity, language, sex, etc. ,and for each individual a Decile score showing the likelihood of recidivism (reoffending). It is known there is bias (Calmon et al., 2017; Rajabi & Garibay, 2022) between ethnicity and Decile score in the sense that Decile score for African-

American group is more likely to be assigned a higher Decile score indicating higher likelihood of recidivism. Therefore, in this experiment, ethnicity is the sensitive feature, and we only keep individuals when the ethnicity is African American and Caucasian. Also, we drop features, such as FirstName, LastName, MiddleName, CASE ID, and DateOfBirth, as people usually do. We convert Decile score as binary variable: "Low Chance of recidivism" when Decile score is less than 5; "High chance of recidivism" for the rest. In a word, sensitive attribute $S = $ ethnicity and the outcome $Y = $ Recidivism Chance.

In what follows, we repeat experiment on COMPAS dataset as we did for Adult dataset in Section 4.1 to show how FDA facilitates the trade-off between fairness (when $\alpha = 0$) and perfect data faithfulness (when $\alpha = 1$) by varying the bias reduction factor $\alpha \in (0, 1)$.

We generate fair synthetic data $\hat{\mathcal{D}}$ by using FDA with various choices of $\alpha$. For different levels of bias reduction factor $\alpha \in [0, 1]$, we report the average of

(1) the empirical estimates of the Wasserstein-2 distance between the synthetic and original data distributions $\hat{W}_2(\mu_{\hat{Y}}, \mu_Y))$ in Figure 5,

(2) the empirical estimates of the unfairness measure $\mathcal{UF}(\hat{\mathcal{D}})$ in the debiased synthetic data in in Figure 6,

(3) the commonly used unfairness measure $|\mathbb{P}(\hat{Y} = 1|S = 1) - \mathbb{P}(\hat{Y} = 1|S = 0)|$ in Figure 7,

across 10 repetitions of experiments.

As we have seen in Section 4.1, it is clear that FDA provides a tuning mechanism on faithfulness and fairness with respect to the bias reduction parameter $\alpha$ comparing with other benchmark method. Furthermore, it is worth noting that the variation of $\hat{W}_2(\mu_{\hat{Y}}, \mu_Y)$ and $\widehat{\mathcal{UF}}(\mathcal{P}_{\hat{\mathcal{D}}})$ for FDA are very small, providing stability of FDA framework.

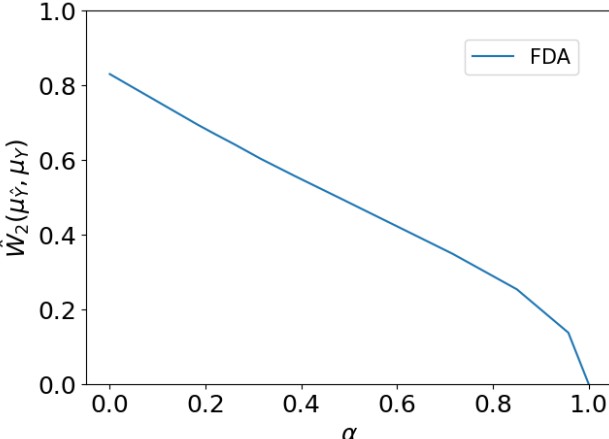

Figure 5: Faithfulness of the generated synthetic dataset by FDA with respect to $\alpha$: $\hat{W}_2(\mu_{\hat{Y}}, \mu_Y)$ ($\downarrow$ more faithful). The shadowed areas along each line represent the variations on 10 repeat ions of experiments.

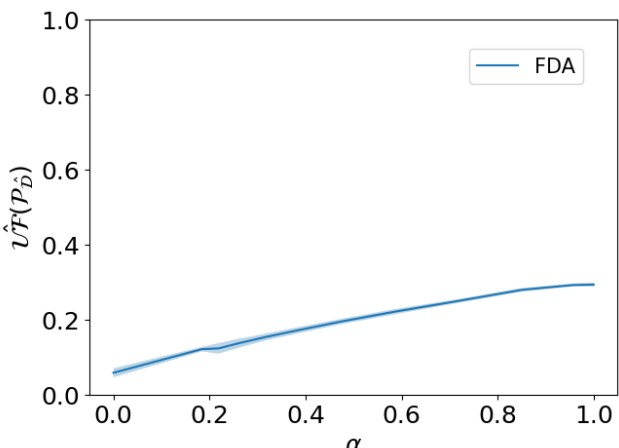

Figure 6: Fairness of the generated synthetic dataset by FDA with respect to $\alpha$: $\widehat{\mathcal{UF}}(\mathcal{P}_{\hat{D}})$ ($\downarrow$ more fair). The shadowed areas along each line represent the variations on 10 repeat ions of experiments.

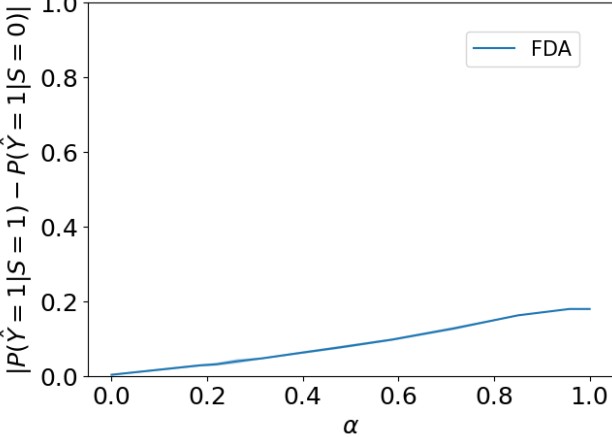

Figure 7: Fairness of the generated synthetic dataset by FDA with respect to $\alpha$: $|\mathbb{P}(\hat{Y} = 1|S = 1) - \mathbb{P}(\hat{Y} = 1|S = 0)|$ ($\downarrow$ more fair). The shadowed areas along each line represent the variations on 10 repeat ions of experiments.

