# OpenReview forum: "FDA: Generating Fair Synthetic Data with Provable Trade-off between Fairness and Faithfulness"
_ICLR.cc/2025/Conference — ICLR 2025 Conference Withdrawn Submission_

### Official Review · Reviewer_EDK8 · 2024-10-21

**Soundness:** 2
**Presentation:** 3
**Contribution:** 1
**Rating:** 3
**Confidence:** 4

**Summary:**

This paper presents an augmentation-based method for generating fair data, applied specifically to the label space.
Not only achieving fairness, but the authors also focus on faithfulness, which is defined by the difference between the distribution of the generated labels and that of the original labels.
The proposed algorithm is flexible, allowing control of the trade-off between faithfulness and fairness.
The modeling used for the augmentation is theoretically sound.
Several experimental results are provided, comparing the proposed method to existing generative methods for fairness.

**Strengths:**

1. The presentation is clear and easy-to-follow.
2. The proposed algorithm can control the trade-off between the faithfulness and fairness.

**Weaknesses:**

1. The major concern is that the proposed method generates only the labels, excluding the input features and sensitive variables. This means that the approach is nothing but only augments the labels of given pairs of (input feature, sensitive variable). Generating the label alone has several critical issues:
- (a) There is a lack of scenarios illustrating the generation of marginal data (i.e., labels only). In what situations would such generated data be needed (i.e., when only labels are generated)? Why is it important and meaningful to generate labels alone?
- (b) The distribution of $(\hat{Y}_i, X_i, S_i)$ would not be similar to the distribution of $(Y_i, X_i, S_i).$
Therefore, the goal of generative modeling (i.e., estimating the distribution) is not achieved.
- (c) Assume that the confidence for the ground-truth response $Y_i$ of a given instance $X_i$ is high.
If $Y_i$ is augmented by a large margin (i.e., $\hat{Y}_i$ and $Y_i$ are significantly different), and a model fits this augmented data well (i.e., the prediction of $X_i$ is very similar to $\hat{Y}_i$).
Then, the prediction performance on $X_i$ will be poor.

2. This problem also affects the experiment parts:
- (a) The baselines (e.g., FairGAN) generate not only the label $Y$ but also the input and sensitive attributes ($X$ and $S$).
Hence, the experimental comparison may not be fair.
- (b) Furthermore, the fairness and faithfulness are only measured in the label space, without considering the generated (joint) distribution (e.g., the similarity between the distribution of generated sample ($\hat{Y}_i, X_i, S_i$) and the original distribution ($Y_i, X_i, S_i$)).

3. In equation (6), if $M \rightarrow \infty,$ it seems that the distribution degenerates to a point mass at 0, not $Y_{i}$. Is equation (6) correct? If I missed some details, please let me know. If not, the proposed distribution is not valid.

4. (Minor) Following the author guide in the ICLR 2025 website
(https://iclr.cc/Conferences/2025/AuthorGuide),
it seems better to add `reproducibility statement’ at the end of the main text.

**Questions:**

1. Consider the following scenario.
First, train a fair model $g$. Then, set $\hat{Y}_i$ by the prediction of $g$ (= $g(X_i)$). In this way, the augmented dataset $(\hat{Y}_i, X_i, S_i)\_{i=1}\^{n}$ would be fair.
What is the main difference between this approach of generating $\hat{Y}_i$ using a trained fair model and the proposed method?

2. (Related to the main concern) What is the main advantage of generating $Y$ only, compared to generating $Y, X, S$ jointly?

3. DECAF generates input data $X$, while $M_{fair}$ generates the label $Y$. In Remark 3.3, the authors claimed that FDA can be combined with DECAF, by using DECAF as $M_{fair}$. How is this combination possible?

---

### Official Review · Reviewer_474G · 2024-10-28

**Soundness:** 2
**Presentation:** 3
**Contribution:** 2
**Rating:** 3
**Confidence:** 3

**Summary:**

This paper proposes a fair synthetic data generation process with provable guarantees over the control between the fairness and faithfulness of the generated data. The proposed method, FDA, jointly uses two models, one to control the fidelity to the original data and the other to control fairness, combined to provide a controllable level of fairness for a given user’s specified fairness level. The study provides theoretical guarantees that fairness and faithfulness can be controlled by the parameter $\alpha$,  also supported by the empirical results.

**Strengths:**

1. The paper is well-written and well-organized.
2. The study introduces a novel fair synthetic data generation method with a controllable balance between fairness and faithfulness to original distribution. This represents a timely problem.
3. Theoretical results are sound and empirical results demonstrate the effectiveness of the proposed method.

**Weaknesses:**

1. Related work should be more thoroughly discussed, and the position of the proposed method with relevant literature can be more clearly provided. In particular, the difference with the work by `Jiang, Bei, et al.` [1] and the line of work on fair representation learning.

2. The method relies on a fair model (Line 203) to generate fair synthetic data with a controllable tradeoff between " faithfulness " and the original target label. The benefit of using the proposed method is unclear, especially when the fair model used already has a controllable tradeoff between fairness and predictive performance. Many state-of-the-art fairness-enhancing algorithms, such as `Agarwal, Alekh, et al.` [2], allow a controllable tradeoff between fairness and accuracy.

3. The experiments are conducted on limited datasets (e.g., Adult Income and Compas). Additional testing on a broader range of datasets, especially in different domains (e.g., image dataset), is necessary to assess the proposed method's generalizability.

4. The proposed method focuses only on statistical parity, limiting its broader usage.

5. Some terminology used in the paper could be misleading and should be better motivated. In particular, the abstract says the proposed method does data augmentation; however, the problem statement (Line 149) shows the proposed method aims to transform the target variable $Y_i$ such that it satisfies the desired fairness property. It is unclear where data augmentation occurs since the new dataset has the same size as the original dataset. The type of data augmentation used should be better clarified.

6. The output dataset is not fully synthetic. The synthetic data obtained by the proposed method differs from the original data only by the target variable, which limits its applicability compared to existing methods such as FairGan and TabFairGan.


[1] Jiang, Bei, et al. "Balancing inferential integrity and disclosure risk via model targeted masking and multiple imputation." Journal of the American Statistical Association 117.537 (2021): 52-66.

[2]Agarwal, Alekh, et al. "A reductions approach to fair classification." International conference on machine learning. PMLR, 2018

**Questions:**

In addition to the weaknesses above, some inconsistencies or clarifications need to be addressed:

1. Since the proposed method does not intend to learn the data distribution from which to sample fair synthetic data, is it fair to compare it with methods that actually learn distributions (i.e., FairGan, TabFairGan)?

2. As the proposed method only “transforms” the target distribution, how does it differ from the work on fair representation learning, which aims to transform the input data into a fairer representation?

3. Regarding the definition of faithfulness used in the paper, can it be equivalent to accuracy when the variable is categorical? In this case, the term “faithfulness to the original data” used throughout the paper is overstated. It should be modified to reflect that it refers to “faithfulness” to the target variable, given that other features remain unchanged.

4. In the experiments (Fig. 2 & 3), how many values of the reduction faction $\alpha$ were used? If the figure does not depict the Pareto front, why is there no fairness value at $\alpha=0.1$, for example?

5. Figure 2 compares the proposed method with other baselines and shows a constant fairness level for all the baselines for different values of $\alpha$. Do the considered baselines use the value $\alpha$? If not, it is better to represent them with straight lines.

6. In Figure 3, adding a state-of-the-art fairness-enhancing method with a controllable tradeoff between fairness and accuracy might strengthen the contribution of the paper.

7. Line 206 says the $f$ can be any fair model; how can the proposed method guarantee that the user-specified fairness level ($\alpha$) is achieved if the fair model cannot reach the expected fairness level, e.g., DP=0?

---

### Official Review · Reviewer_2WsA · 2024-11-01

**Soundness:** 2
**Presentation:** 2
**Contribution:** 1
**Rating:** 3
**Confidence:** 3

**Summary:**

This paper focuses on data-centric methods for mitigating unfairness. It proposes a framework for generating fair data, consisting of two components, M_{faithful} and M_{fair} which ensure that the distribution of the generated data aligns with the target distribution and promotes DP fairness for the downstream task. The author provides recommendations for tuning parameters in practice. In the experimental section, two tabular datasets are used to demonstrate that this method achieves superior performance compared to existing methods in terms of data quality and fairness.

**Strengths:**

1. Fair synthetic data generation methods offer a promising approach to addressing fairness issues. I believe fair synthetic data generation provides a fundamental solution by tackling these issues from a data perspective.

2. The author demonstrates how their method differs from others, clearly stating their contributions.

**Weaknesses:**

**1. Incremental innovation**: After reviewing [1], I personally believe this paper represents incremental work relative to [1].  Concretely, the modeling of $M_{faithful}$ and $M_{fair}$ closely resembles $M_{assoc}$ and $M_{mask}$ from [1], as evidenced by the similarity between equation (4) in this manuscript and Eq (4) from [1], Eq (2) from this manuscript and Eqs (1,2) in [1]. Additionally, Figure 1 in this manuscript and Figure 1 in [1], as well as Algorithm 1 in this manuscript compared to Steps 0-3 from Algorithm 1 in [1], further demonstrate the similarities.

**2. Lack of evaluation** The author limits their comparisons to existing work using only two tabular datasets (UCI Adult and Compass, as detailed in the Appendix), which are not convincing to me. Additionally, the comparison appears unfair to other approaches: The author presents a hyperparameter tuning plot in a figure where only their method is varied, while other methods remain constant. From a perspective of fair data generation, it would be beneficial to include image datasets such as CelebA or Waterbirds to demonstrate the broader applicability and enhance the utility of the work.

**3. Presentation quality** The paper's presentation is suboptimal. The notation is not clear to me, for example, in line 222 $m \in M$, the meaning of $M$ is not introduced beforehand.  Furthermore, the paper does not adequately explain how to replicate the methodologies or the specific functions M_{fair}, M_{faithful}. Details provided in lines 925-931 offer some insight, but they fall short of facilitating replication of the work. I found your code, but this weakness still exists for the presentation purpose.



[1] Jiang, Bei, et al. "Balancing inferential integrity and disclosure risk via model targeted masking and multiple imputation." Journal of the American Statistical Association 117.537 (2021): 52-66.

**Questions:**

1. Would this method achieve a better fairness-DP tradeoff than the post-processing method [1]?
2. Would this method be able to quantify this trade-off?
3. Would this method be beneficial for other fairness metrics like Min-max fairness?

[1] Xian, Ruicheng, Lang Yin, and Han Zhao. "Fair and optimal classification via post-processing." International Conference on Machine Learning. PMLR, 2023.

---

### Official Review · Reviewer_UZ7Z · 2024-11-02

**Soundness:** 3
**Presentation:** 2
**Contribution:** 3
**Rating:** 5
**Confidence:** 4

**Summary:**

This paper introduces a framework called FDA. The framework is designed to produce synthetic data that balances fairness and faithfulness to the original data with provable guarantees. The framework leverages a joint model that incorporates two sub-models:

- M_fair: Focuses on enforcing strict fairness constraints.
- M_faithful: Dedicated to preserving fidelity to the original data.

The trade-off between fairness and faithfulness is controlled by the parameter, α. α=0 means maximum fairness but lower faithfulness, and α=1 means maximal faithfulness but with no fairness modifications.

**Strengths:**

- One strong point is the ability to explicitly balance fairness and faithfulness, which is supported by theoretical proofs.
- The framework can be combined with GAN-based methods (e.g. DECAF). This can be used to improve synthetic data quality in various downstream tasks.
- The paper provides a mathematical proof that fairness can be systematically reduced or increased based on α, with bounds on unfairness for downstream models trained on the synthetic data.

**Weaknesses:**

- The model relies on some hyperparameters (σ2 and M). The authors claim that it takes 1.3 seconds on the Adult dataset
with n = 63,000 in the training set. Using a Core i5-12600K CPU (16 threads). Does the code run multi-threaded? I assume it does so we have around 330us / instance and since we have 14 features we have: around 23us (1.3 sec / (63000/16) / 14). If we take this to an image dataset with 784 features the method may face scalability issues in large or high-dimensional datasets where traditional methods are more efficient.
- While the paper’s theoretical contributions are strong, empirical examples are limited to benchmarks like the UCI Adult / COMPAS dataset. What about image datasets etc?
- While FDA shows improved performance over FairGAN and DECAF, a more extensive comparison with other fairness-focused synthetic data generation methods would highlight FDA’s advantages and limitations better.

**Questions:**

- Could you provide practical guidelines for setting the hyperparameters σ2 and M to achieve optimal trade-offs? Are there heuristics that practitioners could follow?
- How does the FDA framework perform with high-dimensional datasets? Are there computational shortcuts or approximations that could be applied to maintain efficiency?
- The paper uses Demographic Parity (DP) as the primary fairness measure. How would FDA perform with alternative fairness measures, such as Equalized Odds or Predictive Parity?

---

### Note · Authors · 2024-11-17

I have read and agree with the venue's withdrawal policy on behalf of myself and my co-authors.